# Innovative Structural Systems for Timber Buildings: A Comprehensive Review of Contemporary Solutions

Miroslav Premrov [1] and Vesna Žegarac Leskovar [2],*

1    Department for Civil Engineering, Faculty of Civil Engineering, Transportation Engineering and Architecture, University of Maribor, 2000 Maribor, Slovenia; miroslav.premrov@um.si

2    Department for Architecture, Faculty of Civil Engineering, Transportation Engineering and Architecture, University of Maribor, 2000 Maribor, Slovenia

\*    Correspondence: vesna.zegarac@um.si; Tel.: +386-4061-6090

**Abstract:** The remarkable development of timber construction technologies in recent decades has led to an increase in the number of timber buildings worldwide, including multi-storey buildings. The design of timber buildings, especially those of greater height, is relatively demanding and, even in the context of architectural expression, has certain constraints due to the specific structural and physical properties of this material. Thus, it is important for designers to have an overview of existing timber structural systems and their specificities to be able to make the right design decisions during the design process. Unfortunately, there is a lack of scientific literature that systematically addresses the essential features of contemporary timber structural systems. Within the aims of this paper to provide the systematic review of contemporary structural systems, both the scientific and professional literature are comprehensively reviewed. This paper presents a systematic classification and description of the following structural systems of timber buildings: all-timber and hybrid timber structural systems with an additional description of constituent structural elements, while examples of completed multi-storey timber buildings are also given for each structural system. The findings provide a broader view of the knowledge of contemporary solutions of timber structural systems and their application, thus representing a novelty in the field of timber construction review.

**Keywords:** timber structural systems; hybrid timber structural system; timber-concrete composite; timber-steel composite; timber-timber composite; timber structural elements; timber buildings; multi-storey timber buildings

## 1. Introduction

In recent decades, we have been witnessing a notable development of timber construction. There are several reasons why wood as a building material is enjoying a resurgence in the construction industry. Contemporary construction techniques, newly developed pure and hybrid timber structural systems, and newly developed timber composite structural elements facilitate the construction of higher, taller, and more challenging buildings in terms of architectural design expression. Another argument that supports the increasing interest in bio-based building materials is climate change. Wood, because of its natural origin, its ability to sequester carbon, its recyclability, and its natural decomposition ability, meets the requirements of environmental protection far better than any other building material. Moreover, wood is an easy-to-process material, and much of the construction process can be carried out as prefabrication in the factory. In addition to the previously listed reasons, wood has excellent structural features. Its compressive strength is almost equal to that of concrete, but its tensile strength is significantly higher [1–3]. Also, an important advantage over concrete is its much lower weight [1–3]. From the economic point of view, the primary costs compared to common standard solutions may be somewhat higher than those for conventional constructions, while in terms of overall cost-effectiveness,

contemporary timber construction can outperform conventional construction [4]. However, the design of timber buildings is slightly more challenging due to the specific structural and physical properties of this material. Challenges to be highlighted include its sensitivity to moisture, anisotropy, and low stiffness [5]. Still, there are certain limitations that influence the architectural expression of timber buildings to a certain extent. The complexity of building design is reflected in the selection of a suitable architectural design concept and of a suitable structural system, and the energy efficiency concept, which all strongly depend on the specific features of the location, particularly climate conditions, wind exposure, and seismic hazard [3]. To make proper design decisions for timber buildings, it is important to obtain an overview of all existing timber structural systems and timber composite elements, and to understand their capacities and characteristics.

Currently, there exists a large body of literature on timber construction, but it is not possible to find articles in the scientific literature comprehensively covering all the main structural systems that are either purely timber or based on timber construction. The present review paper is therefore a novelty from the point of view of providing a systematic overview of the state of the art in this field. In addition, it can also serve researchers as a kind of encyclopaedia of modern timber construction systems and planners in the selection of alternative existing solutions.

To this end, the current paper contains an overview of existing timber building structural systems, pure and hybrid, and timber-based structural elements. The data are given in two scales: at the structural system level and at the element level, while the scale of the connecting elements is not considered in this paper. Additionally, the application of the presented structural systems and structural elements in high-rise timber construction is shown.

## 2. Methodology of Literature Research

The aim of this study is to conduct a systematic analysis of scientific literature that focuses on existing contemporary timber-based structural systems, particularly in terms of their specifics and application in buildings. The methodology used in the present literature review was designed following the Preferred Reporting Items for Systematic Reviews and Meta-Analyses (PRISMA) guidelines [6], while expert literature relevant for the study topic was additionally reviewed and used in this study.

### 2.1. Literature Research Based on Preferred Reporting Items for Systematic Reviews and Meta-Analyses (PRISMA) Guidelines

The databases used were Scopus, ScienceDirect, and Directory of Open Access Journals (DOAJ) (Table 1). Scopus was selected as one of the two largest bibliographic databases that covers scholarly literature from almost any discipline. In addition, Science Direct was selected since it covers a large number of indexed scientific works in the fields of engineering, while DOAJ was chosen as an open-access academic database. Initially, in the first step of the literature search, the records were identified from selected databases using the query title-abstract-keyword.

In order to focus on the literature addressing timber-based structural systems, the selected set of keywords consisted of the following:

- Set 1: timber structural system;
- Set 2: hybrid timber structural system;
- Set 3: timber-concrete composite/element;
- Set 4: timber-steel composite/element;
- Set 5: multi-storey timber/wood buildings.

Additionally, the following inclusion criteria were considered to filter the review collection (Table 2):

- Original research scientific articles, systematic literature review articles, monographs, books, book chapters with accessible full text and thematic relevance to our goal were included;

- Only works published in 2000 and after were considered, since we were interested in contemporary solutions and the studies on this topic published before 2000 are rare;
- Scientific literature was mainly published in English, while for publications in other languages, it was considered that the title, abstract, and keywords data, in case of being translated into English, do not provide enough information for a relevant analysis of the content, and this literature also represents a smaller percentage compared to the English-language literature share.

**Table 1.** Specification of scientific databases and the number of literature records under observation.

| Databases | Keywords | No. of Records | No. of Selected Records | No. of Records Accessed for Eligibility | References of Records Accessed for Eligibility |
|---|---|---|---|---|---|
| ScienceDirect | Timber structural system | 492 | 39 | 19 | [7–25] |
| | Hybrid timber structural system | 67 | 47 | 3 | [12,15,26] |
| | Timber-concrete composite | 275 | 115 | 8 | [27–34] |
| | Timber-steel composite | 225 | 45 | 5 | [35–39] |
| | Timber-timber composite | 204 | 11 | 3 | [40–42] |
| | Multi-storey timber buildings | 79 | 55 | 4 | [5,43–45] |
| Scopus | Timber structural system | 1985 | 28 | 17 | [8–16,19–25,46] |
| | Hybrid timber structural system | 151 | 10 | 6 | [12,15,26,47–49] |
| | Timber-concrete composite | 356 | 32 | 9 | [28–34,50,51] |
| | Timber-steel composite | 63 | 9 | 4 | [36,38,39,49] |
| | Timber-timber composite | 20 | 5 | 5 | [40,41,47,52,53] |
| | Multi-storey timber buildings | 175 | 9 | 5 | [5,43–45,54] |
| DOAJ | Timber structural system | 116 | 8 | 0 | 0 |
| | Hybrid timber structural system | 9 | 3 | 3 | [55–57] |
| | Timber-concrete composite | 86 | 38 | 0 | 0 |
| | Timber-steel composite | 58 | 15 | 3 | [55,57,58] |
| | Timber-timber composite | 215 | 2 | 0 | 0 |
| | Multi-storey timber buildings | 14 | 11 | 4 | [54,59–61] |
| Total | | 4532 | 482 | 98 | Final No. of Records 56 (excluding 42 duplicate Records) |

**Table 2.** Inclusion and exclusion criteria.

| Inclusion Criteria | Exclusion Criteria |
|---|---|
| Published in English | Not published in English |
| Access to full text | Access only to abstract or bibliographic data |
| Original research scientific articles, systematic literature review articles, monographs, books, book chapters | Conference abstracts, book reviews, conference info, correspondence, editorials, mini reviews, product reviews, short communication |
| Papers published in 2000 and later | Papers published prior to 2000 |

Table 2 shows the inclusion and exclusion criteria on the basis of which the final scientific literature selection was conducted.

Considering the above inclusion criteria (Table 2), 4532 records were identified in the first step, of which 4050 records that were not accessible in the full-text version were removed. The initial screening excluded 384 records that were not directly related to our review. After this screening, 98 records were reviewed for eligibility. The second screening excluded 42 duplicate records. Finally, 56 records met our selection criteria and were further discussed in Sections 3 and 4 of the current study (Table 1, Figure 1).

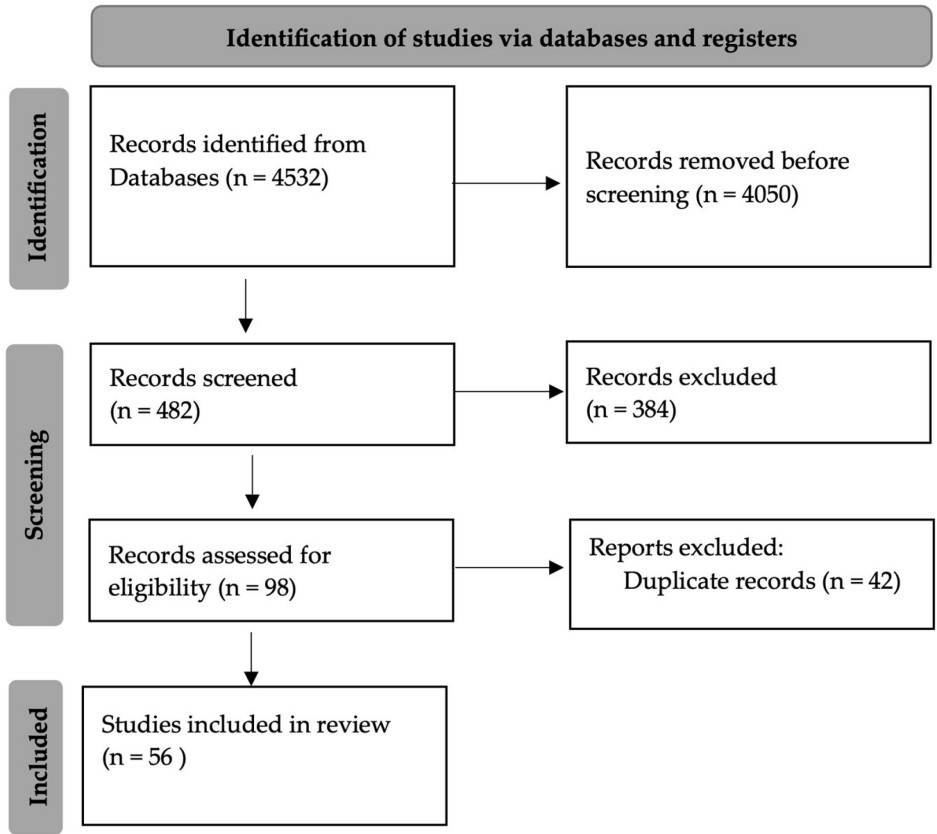

**Figure 1.** PRISMA 2020 flow diagram for new systematic reviews.

### 2.2. Additional Literature Search Using Expert Sources

In addition to the systematic review of the scientific literature, technical expert literature covering the field of timber structural systems was also reviewed and used in this study. This paper included 30 relevant expert literature sources, such as books and expert papers, web-accessible expert documents, standards, and one doctoral dissertation, which were not listed in a systematic review of scientific literature. Despite the non-scientific nature of these sources, we found them to be extremely relevant to this paper, as they addressed the topics of timber structural systems and timber-based composite elements. A total of 86 sources were used for this paper.

### 2.3. The Aim of the Selected Literature Review

As an upgrade of existing studies, most of which address just individual timber structural systems or timber structural elements, or only certain types of timber construction, the current review paper aims to include a systematic classification of timber structural systems according to their load-bearing function, followed by graphical and textual interpretation of individual systems and their corresponding structural elements used in contemporary timber building construction. This paper does not only focus on individual systems or structural elements, but systematically summarises the latest research results in the field of structural systems in timber construction. Thus, this study is a significant scientific contribution in the context of the transparency of existing systems, their main associated load-bearing structural elements, and the possibilities of their applications in modern timber construction.

### 3. Structural Systems of Timber Buildings

Selecting a load-bearing timber construction system depends primarily on architectural demands, with the orientation, location, and the purpose of a building being of no lesser importance. Load-bearing timber construction systems differ from each other in the

technological aspect (conventional and prefabricated), the appearance of the structure, and the approach to planning and designing a particular system.

To select a suitable solution, it is necessary to obtain an overview of the existing systems, which can be divided into the following two categories:

- Timber structural systems, in which only timber is used for the main load-bearing components;
- Hybrid timber-based structural systems, in which structural components made of other structural materials, like concrete and/or steel, are additionally used for load-bearing structures.

Initially, the main timber structural systems are presented in the following subsection.

As presented in [62], load-bearing parts of timber buildings can be classified into six major structural systems (Figure 2):

- Log construction (non-prefabricated);
- Solid timber construction (prefabricated);
- Timber frame construction (non-prefabricated);
- Frame construction (non-prefabricated);
- Balloon and platform frame construction (non-prefabricated);
- Frame-panel construction (prefabricated).

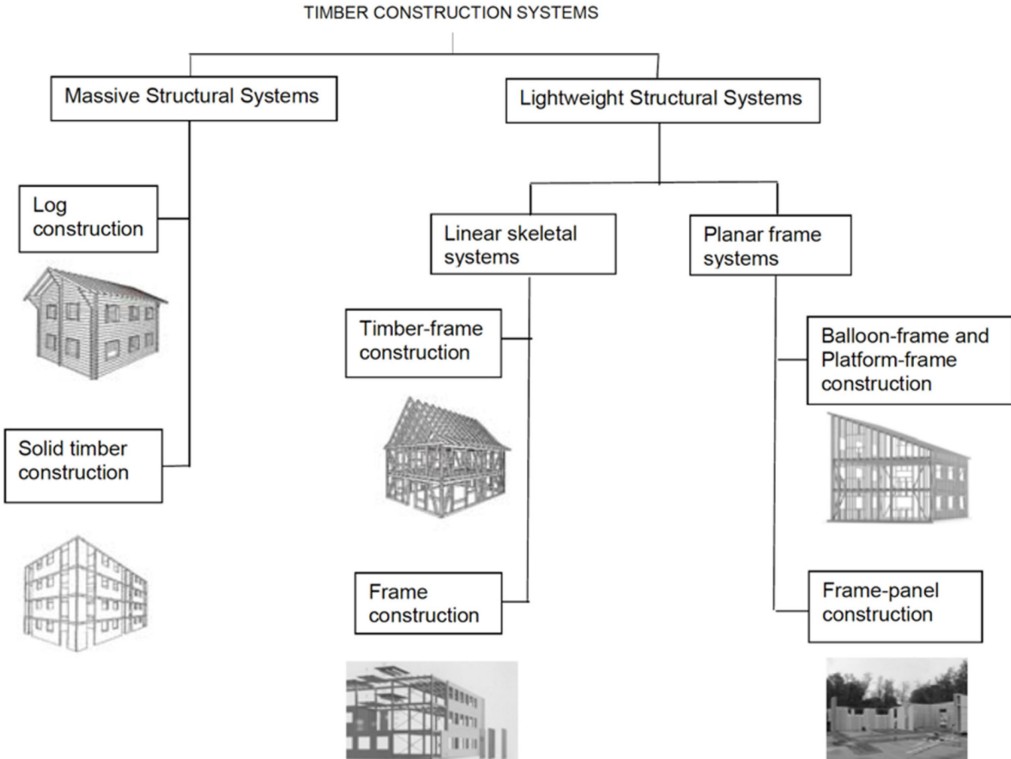

**Figure 2.** Classification of timber structural systems according to their load-bearing functions [3].

Log construction and solid timber construction can also be classified as massive structural systems, since all load-bearing wall and floor elements consist of massive solid elements. Log construction is the most traditional type of timber construction used in many countries in the world, especially in areas with cold climate conditions, like Scandinavia, in the Alps, and in the mountainous regions of Central Europe. Log construction includes the most massive type of timber structures and usually the most expensive type of timber construction as well. Therefore, log construction usually plays a significant role, especially for local inhabitants' houses. However, due to certain structural and cost limitations, it is not appropriate for mid-rise multi-storey timber buildings.

Solid timber construction as a contemporary massive structural system consists of prefabricated massive panels made from cross-laminated timber (CLT) wall or floor elements, which eliminate a strong anisotropy of wood used in conventional old log construction. Solid cross-laminated timber panels are made from three to eleven fir plies glued together crosswise. The resulting homogeneous, dimensionally stable, and rigid component can be produced in sizes up to 4.8 m × 20 m. Available thicknesses depending on the number of plies range from 50 mm to 300 mm [63]. The application of CLT panel elements opens many new perspectives in designing contemporary multi-storey timber buildings [7,43,62]. Before the hybrid structural combination of various timber systems was used, the tallest timber buildings had been built in the CLT structural system with the maximum height of up to ten storeys. The ten-storey building Forte (Melbourne) built in 2012 with a total height of 32.7 metres was regarded at that time as the tallest timber building in the world. Another interesting example is the nine-storey building Via Cenni (Milan) constructed in 2013 as a demonstration case of the highest CLT building in the world constructed in an active seismic area. It is recognised as the first high-rise full CLT timber building located in an active seismic area [64].

Other construction systems are classified as lightweight structural systems. According to their load-bearing function, they can be subdivided into conventional linear skeletal systems, in which all the loads are transmitted with linear bearing elements and planar frame systems, also known as light timber-framed (LTF) structures, in which sheathing boards take over horizontal loads and consequently structural wall elements can be treated as two-dimensional planar components [3].

The prefabricated frame-panel construction originates from the Scandinavian-American conventional non-prefabricated construction methods, i.e., balloon frame and platform frame construction types, which are assembled on-site [65]. The advantages of the frame-panel construction systems over the said traditional timber frame construction systems were first noticed at the beginning of the 1980s and significantly contributed to the development of timber construction. One of the main advantages of all LTF wall components is that a large amount of thermal insulation can be placed between the timber frame structural elements, which is not possible in the case of CLT elements. On the other hand, solid timber construction (CLT) proves higher lateral resistance and stiffness, and also better acoustic and fire resistances.

It is important to point out that only solid timber construction (CLT) and frame-panel construction are prefabricated; other structural systems are conventional and usually built on-site. Therefore, we will mainly focus on these two prefabricated systems to analyse the main characteristics of sustainable prefabricated timber construction.

Table 3 provides an overview of the basic construction features of all six main timber structural systems.

**Table 3.** Basic structural characteristics of main timber structural systems.

| Main Timber Structural Systems | | | | |
|:---:|:---:|:---:|:---:|:---:|
| Type | Massive | Lightweight | Construction Process | Use in Contemporary Construction |
| LOG | ✔ | | on-site | |
| SOLID TIMBER (CLT) | ✔ | | prefabricated | ✔ |
| TIMBER FRAME | | ✔ | on-site | |
| FRAME | | ✔ | on-site | ✔ |
| BALLOON AND PLATFORM FRAME | | ✔ | on-site | |
| FRAME-PANEL | | ✔ | prefabricated | ✔ |

As seen in Table 3, three building systems are predominantly used in contemporary construction, i.e., solid timber, frame, and frame-panel. Of these, however, the two most

commonly used are frame-panel and CLT system, both as prefabricated structural systems. Conventional light timber-framed construction, balloon-frame or platform-frame, is mainly used in North America, New Zealand, and Europe. Especially in North America, most housing and commercial structures used wood as the main structural material until the 20th century [66].

For the purpose of further detailed description of the said three contemporary timber structural systems, Table 4 provides a graphical presentation of their structural elements (components).

**Table 4.** All-timber structural elements according to their usages in main timber structural systems. (Drawings adapted after [4]).

| | | All-Timber Structural Elements—Division to Structural Systems | | |
|---|---|---|---|---|
| | **Element** | **Massive** | **Lightweight** | |
| | | **Solid Timber (CLT Panel)** | **Linear Skeletal** | **Planar Frame (Frame-Panel)** |
| 1D | COLUMN BEAM DIAGONAL | / |  | / |
| 2D | FLOOR CEILING |  | Floor/ceiling structure consists of classic timber ceiling joists—not necessarily produced as prefabricated structural elements. |  |
| 2D | WALL |  |  |  |
| 2D | ROOF |  | Roof structure consists of classic timber rafters—not produced as prefabricated structural elements. | Roof structure consists of classic timber rafters—not produced as prefabricated structural elements. |
| 3D | MODULAR BOX |  | / |  |

In Table 4, the structural elements are provided according to the associated structural system and their dimensionality, i.e., one-, two- and three-dimensional. As presented in the solid timber (CLT panel) structural system [46], all structural components (floor, wall, and roof elements) are full and therefore regarded as two-dimensional (2D) structural elements. There are many different types of massive panel structural floor and wall elements used as modular elements, usually built in standard dimensions. Prefabricated floor and wall elements consist of cross-laminated timber boards which are laminated perpendicularly to each other. Consequently, the anisotropy effect of such timber elements can be neglected, opening many new perspectives on designing taller timber buildings, mainly mid-rise buildings (between four and ten storeys). The bracing system against horizontal load impact can consist only of prefabricated CLT wall elements. Despite the lack of experimental studies, the effect of horizontal rigid structural (floor) diaphragms can be achieved by using only prefabricated CLT floor elements of proper thicknesses [67]. To reduce the erection time of multi-storey timber buildings, 3D modular construction was

developed. Three-dimensional (3D) timber modules are composed of two-dimensional (2D) elements, such as walls, floors, and roofs.

In the linear skeletal system, all load-bearing structural elements are linear (one-dimensional) made from solid or, in the case of larger cross-sections, with glue-laminated timber. The sheathing boards or infills with thermal-insulating material do not contribute to the resistance of wall elements. All vertical and horizontal loads are therefore transmitted via vertical (studs), horizontal (beams), and diagonal timber components. As schematically presented in the figure, the bracing wall system against a horizontal load impact can be achieved exclusively by using additional diagonal elements, which can be made in solid or glue-laminated timber (conventional timber frame construction) or in steel (contemporary frame construction). Due to the extremely high slenderness of the steel diagonals, only steel diagonals in tension can be considered as load-bearing members. Therefore, if steel diagonals are used, they are placed in two perpendicular directions to assume the horizontal load (wind, earthquake) acting from two possible opposite directions.

The main difference in the structural behaviours of linear skeletal and frame-panel structural wall components is that the bracing resistance in a frame-panel wall is achieved only by using sheathing boards, which are mechanically connected to the timber frame components (studs and girders). Therefore, a frame-panel wall element is a composite structural element composed of linear timber elements to assume the vertical load actions and two-dimensional sheathing panels to assume the horizontal load impact. Therefore, frame-panel components are regarded as 2D structural elements. It is important to point out that the lateral resistance of a wall element essentially depends on the type of the sheathing materials (OSB or fibre-plaster boards) and the type of connections between sheathing boards and timber frame elements [8–10]. Because the overall lateral resistance is mainly achieved with sheathing boards and the type of connection, not by timber frame elements, it is important to mention that there is usually no need to insert additional diagonal elements in the frame-panel wall construction. They can be used only if wall elements are strengthened to increase the racking resistance of bracing wall elements in taller (for example, four-storey) frame-panel timber buildings located at a heavily windy or seismic location, and when the diagonal tensile resistances of sheathing boards in timber buildings constructed in the frame-panel structural system cannot be achieved by the type and thickness of the sheathing material [68]. Therefore, in such cases, additional special diagonal elements like timber, steel, or CFRP diagonals must be inserted and fixed to the timber frame to increase the horizontal resistance of the whole bracing wall element [8,11].

## 4. Hybrid Timber Structural Systems

For timber buildings with more demanding boundary conditions in terms of stability, the load-bearing capacity, the desired height and spans, vibrations, fire, and sound resistance, and of architectural, ecological, and energy approaches, more suitable solutions can be found with hybrid timber structural systems. There are two different alternatives of hybrid timber buildings, which may also be used for wall and floor elements:

- Combining various timber structural systems made of timber components exclusively described above (mostly LTF and CLT structural elements or CLT and glued-laminated frame elements) is called an all-timber hybrid structural system;
- Combining timber with another structural load-bearing material (mainly with concrete or steel) is called a hybrid timber-based structural system [26].

Both alternatives are graphically presented in Table 5 with all possible different combinations with the implementation of various structural systems and their positions in the building, and with different structural materials (concrete and/or steel) used as structural strengthening elements to increase the load-bearing capacity and stiffness of a timber-based building.

**Table 5.** Graphic presentation of hybrid timber structural systems.

| Hybrid Timber Structural Systems | | | |
|---|---|---|---|
| **All-Timber Hybrid Structural Systems** | **Hybrid Timber-Based Structural Systems** | | |
| **Timber-Timber** | **Timber-Concrete** | **Timber-Steel** | **Timber-Concrete-Steel** |
| CLT core + skeletal frame construction on the envelope of the building, example is the Treet building in Bergen [69]<br><br>Column<br>Column to column connection<br>Top plate<br>Solid Part<br>Bottom plate<br>Column to column connection<br><br>[44] | (a) (b) (c)<br><br>(a) Concrete podium<br>(b) Concrete core<br>(c) Concrete podium and core<br><br>(d)<br><br>(d) Timber wall + concrete podium + TCC floor<br><br>(drawings adapted from [70]) | External reinforcing with a special steel frame structure (drawings adapted from [70]) | Concrete core + special steel frame structure (drawings adapted from [70]) |

## 4.1. All-Timber Hybrid Structural Systems

As mentioned in Section 3, a comparison shows many advantages and disadvantages of both the LTF and CLT structural systems also considering structural and building energy-efficiency concepts. Such a combination of all-timber hybrid elements can be performed on the level of load-bearing structural elements (floors and walls) described in Section 4.1.1, and further on the level of structural systems discussed in Section 3.

### 4.1.1. Composite All-Timber Structural Elements

In addition to the use of structural elements that belong exclusively to the described main timber structural systems, as shown in Table 4, modern timber constructions contain structural elements that are composed of system-specific members and combine the advantages of different timber structural components. These structural components are called composite elements, while some researchers also call them hybrid elements. For a better illustration, some examples of all-timber composite structural elements are graphically presented for floor and wall elements in Table 6.

**Table 6.** Graphic presentation of all-timber composite structural elements.

| All-Timber Composite Structural Elements |
|---|

**(a) CLT slab with timber beams (ribbed CLT) [40,41,47,52]**

FLOOR
CEILING

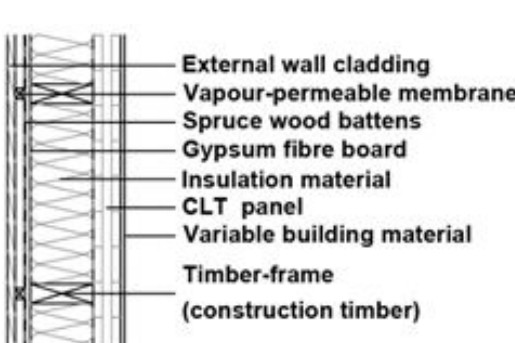

**(a) Combination of CLT and LTF components**          **(b) Hybrid timber frame (HTF) [12]**

2D

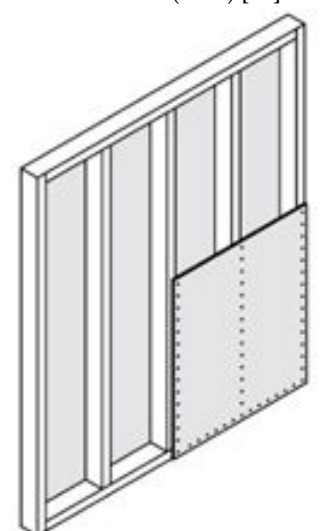

WALL

Linear CLT frame elements + OSB sheathing boards

**(c) Combination of glued-laminated timber (GLT) frame elements with a cross-laminated timber (CLT) shear panel as an infill [13]**

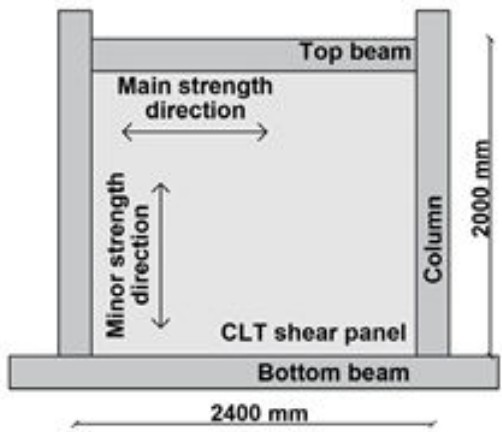

Composite Floor Elements

Regarding structural elements, selecting the type of a prefabricated floor element primarily depends on the floor span and the purpose of the floor area. In residential buildings, where floor spans exceed six metres, the usual choice is a cross-laminated (CLT) floor system due to its higher bending resistance. However, sometimes, the CLT deck is used as a strengthening element placed above the solid timber or glulam joists [40,47,52]. This application is also preferred for the floor vibrational behaviour, as it also increases the floor bending stiffness perpendicular to the joist direction. Floor spans up to ten metres can be achieved with this composite all-timber floor element. CLT slabs are screwed to timber joists even without predrilling when self-tapping screws are used in soft timber. In the case of structural renovation, an additional advantage of this strengthening floor solution with CLT slabs is very low additional mass, which is especially important when the considered building has walls and foundations of questionable strength.

Composite Wall Elements

As described above, the CLT panels provide higher stiffness and load-bearing capacity. On the other hand, light timber frame solutions are better from the aspect of thermal insulation. Therefore, it is sometimes convenient to combine both timber components in one hybrid all-timber wall element, in which usually a CLT panel as the main vertical load-resisting component is placed on the inner side of the building and the light timber-framed (LTF) component as the main thermal insulating vertical component is placed on the external side of the building. As seen in Table 6, a new hybrid structural wall system, consisting of LTF and CLT prefabricated wall elements and denoted as hybrid timber-frame (HTF), takes advantage of the strong prefabrication, reduced weight of light frame timber systems (LTF), and the excellent strength properties of the cross-laminated timber (CLT) panels. Specifically, solid timber members typically used in the structural elements of light frame systems are replaced by CLT linear elements. Therefore, the HTF wall system is an evolution of the LFT wall with an increased lateral resistance, in which the outer timber members of the structural frame (i.e., top and bottom beams, and outer studs) are made with CLT elements [12].

Another possible composite combination of CLT elements with another one made from timber is a newly developed timber structural wall system with a combination of glued-laminated timber (GLT) with cross-laminated timber (CLT). It has recently been proposed by the study in [13], highlighting the important influence on the design of timber frame-shear wall structure. The GLT main frame structure is infilled with CLT panel wall elements, aiming to improve the lateral performance of timber and satisfy the seismic demands. It has been shown, for example, that the load-bearing capacity increased by 95–127.5% when the thickness of the three-layer CLT shear wall increases from 30 mm to 105 mm. The structural–non-structural interaction effects of non-structural partition walls and post-tensioned CLT rocking walls in mass timber buildings were further evaluated in a parametric study [14].

4.1.2. Examples of Already Erected Tall Buildings in All-Timber Hybrid Structural Systems

It is important to point out that massive panel and frame-panel structural systems are, from the technological aspect, prefabricated, and wall and floor elements are produced as typological two-dimensional panels. For the construction of both systems, the erection time is practically identical and very short, which is one of the most important technological parameters in combining these two different structural systems. The goal is to integrate all benefits of the established structural systems, such as the massive panel system [3] with basic load-bearing elements made from cross-laminated plate panel elements (CLT), and the frame-panel structural system, in which the structural stability is based on the composite action of timber frame elements (studs and girders) and sheathing boards (LTF system). Such an example of an erected building with a combination of CLT and LTF structural systems is an eight-storey apartment prefabricated timber building Limnologen

complex in Växjö (Sweden), consisting of eight storeys above the ground with the first storey in concrete and seven storeys in timber. Constructed in 2009, the building was selected as the highest timber building in Sweden built at that time [71]. It is especially important from the structural point of view that the floor plan design of these buildings is asymmetric. Consequently, slightly important torsional actions can apply primarily to envelope wall elements due to the wind load impact. Therefore, a combination of massive panel (CLT) wall elements as envelope structural elements with higher racking resistance and stiffness and light timber-framed (LTF) elements as less load-resisting for internal walls is used to withstand the horizontal load impact [59]. Due to greater lateral forces acting on the building envelope, it is important that CLT wall elements are placed on the building envelope to assume most of the torsional effects due to the buildings' asymmetric plan form, and the LTF elements are used for internal walls, in which this torsional influence is lower.

The example of a structural hybrid combination of CLT elements with the glulam frame elements is the Treet building in Bergen presented in Table 5. The 14-storey timber building completed in 2015 was selected as the highest timber building in the world at that time. The structural system consisted of glued-laminated frame elements and a CLT core [69]. Additionally, even considering only the wind load impact as decisive, the building floor plan is very compact and rectangular without any special forms. It can be predicted that such a 14-storey timber building cannot be constructed in more seismically active areas or can resist the seismic load impact only by using additional special reinforced concrete cores. In the latter, the whole horizontal load impact is assumed only to be sustained by glulam frame elements and there is no contribution of the CLT core to the overall lateral stability of the whole building. The CLT core is a vertical load-bearing component only for elevators and staircases.

In contradiction, the 18-storey Mjøstårnet building in Brumunddal erected in 2019 is constructed in the hybrid CLT-timber frame structural system to withstand the whole horizontal load impact. With the height of 85.4 metres, the building has been verified as the world's tallest timber building by the Council on Tall Buildings and Urban Habitat [72].

Another innovative structural timber system applicable to mid-rise and high-rise timber buildings with irregular floor shapes was developed [44]. The building system made from timber elements exclusively combines the strengths of massive timber constructions and hollow box systems to achieve a point-supported slab with multi-directional spans. It consists of full timber layups in highly stressed areas around the column head and hollow build-ups in open span areas. The developed slab system combines hardwood and softwood materials in a sandwich construction. Furthermore, the developed building system is demonstrated in the design and construction of the ITECH Campus Lab. Finally, the integrative structural design of a timber-fibre hybrid building system fabricated through coreless filament winding project, Maison Fibre, goes one step further and adapts the fabrication to a hybrid structure combining fibre-polymer composites (FPC) with laminated veneer lumber (LVL) to allow for walkability [15].

### 4.2. Hybrid Timber-Based Structural Systems (Combining Timber with Another Structural Material)

One of the major disadvantages of lightweight timber buildings beyond a relatively bad acoustic performance is a low thermal capacity, which can have an important negative impact on the energy demand of such buildings. Therefore, a feasible solution for mitigating the overheating of timber buildings while taking advantage of their low environmental impacts is adding heavyweight materials to compensate for the low thermal capacity of these constructions [16]. Different researchers have studied the effect of adding thermal mass to lightweight timber buildings to improve the thermal capacity of buildings [17,18,73]. In such a case, an addition of massive and thermal capacitive materials (concrete, masonry, stone, etc.) seems to be a good solution. There are also some structural disadvantages of load-bearing timber elements, such as a low value of the modulus of elasticity, which results

in many limitations to the height of timber buildings or floor spans due to the Eurocode serviceability limit state (SLS) criteria [74]. In this sense, it is recommended to combine timber with another structural material like concrete or steel to increase the load-bearing capacity and stiffness of structural elements. On the other hand, it is sometimes important to increase the ductility of the whole structure. To this end, a hybrid combination with steel elements is highly recommended. However, it is important to emphasise that, in this case, thermal capacity will not improve, as it is in a hybrid combination of timber with concrete components. Because another structural material is usually used as the reinforcing material of timber elements or structural systems to improve the behaviour of a timber construction, we usually call these structural systems timber-based systems.

In [48], the researchers presented a review of the seismic behaviour of timber-based hybrid buildings, summarising most of the hybrid timber buildings constructed prior to 2017, and research on the topic prior to 2019. According to the authors, hybrid timber-based structural systems can be grouped into five categories, in which timber is coupled with the following:

- Reinforced concrete;
- Masonry walls;
- Traditional steel framing dissipating steel braces;
- Seismic protection devices;
- Other less common hybrid structural systems (like timber-glass structures).

### 4.2.1. Composite Timber-Based Structural Elements/Components

In the case of hybrid timber-based structural systems, structural elements that are composed of system-specific members or of different structural materials can be applied to contemporary construction in addition to structural elements that belong exclusively to a particular structural system or are composed of one structural material only. If used predominately in timber construction, these elements are usually called composite timber-based structural elements. Some of the possible structural composite combinations with reinforced concrete and steel elements described more in-depth below are graphically presented in Table 7.

**Table 7.** Graphic presentation of timber-based composite structural elements.

| | | Timber-Concrete | Timber-Steel |
|---|---|---|---|
| 1D | COLUMN BEAM DIAGONAL | Presented as 2D TCC floor element | Columns [35]<br>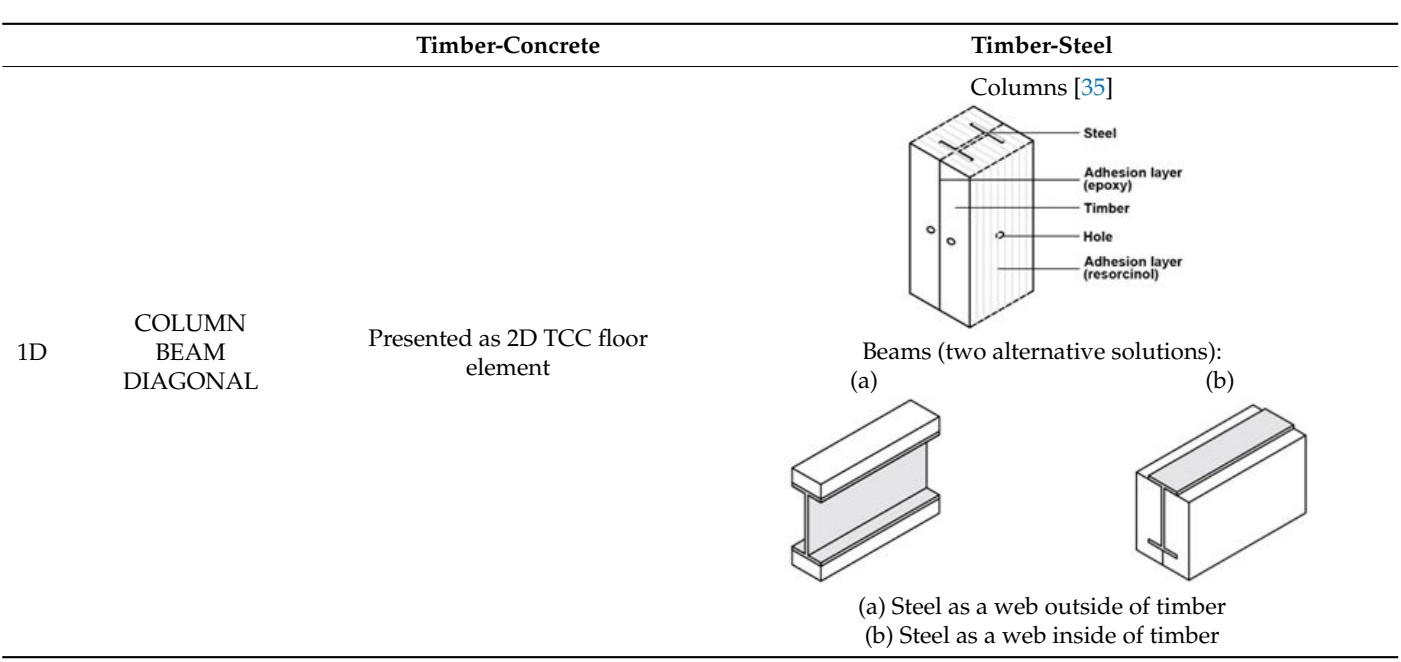<br>Beams (two alternative solutions):<br>(a) (b)<br>(a) Steel as a web outside of timber<br>(b) Steel as a web inside of timber |

**Table 7.** *Cont.*

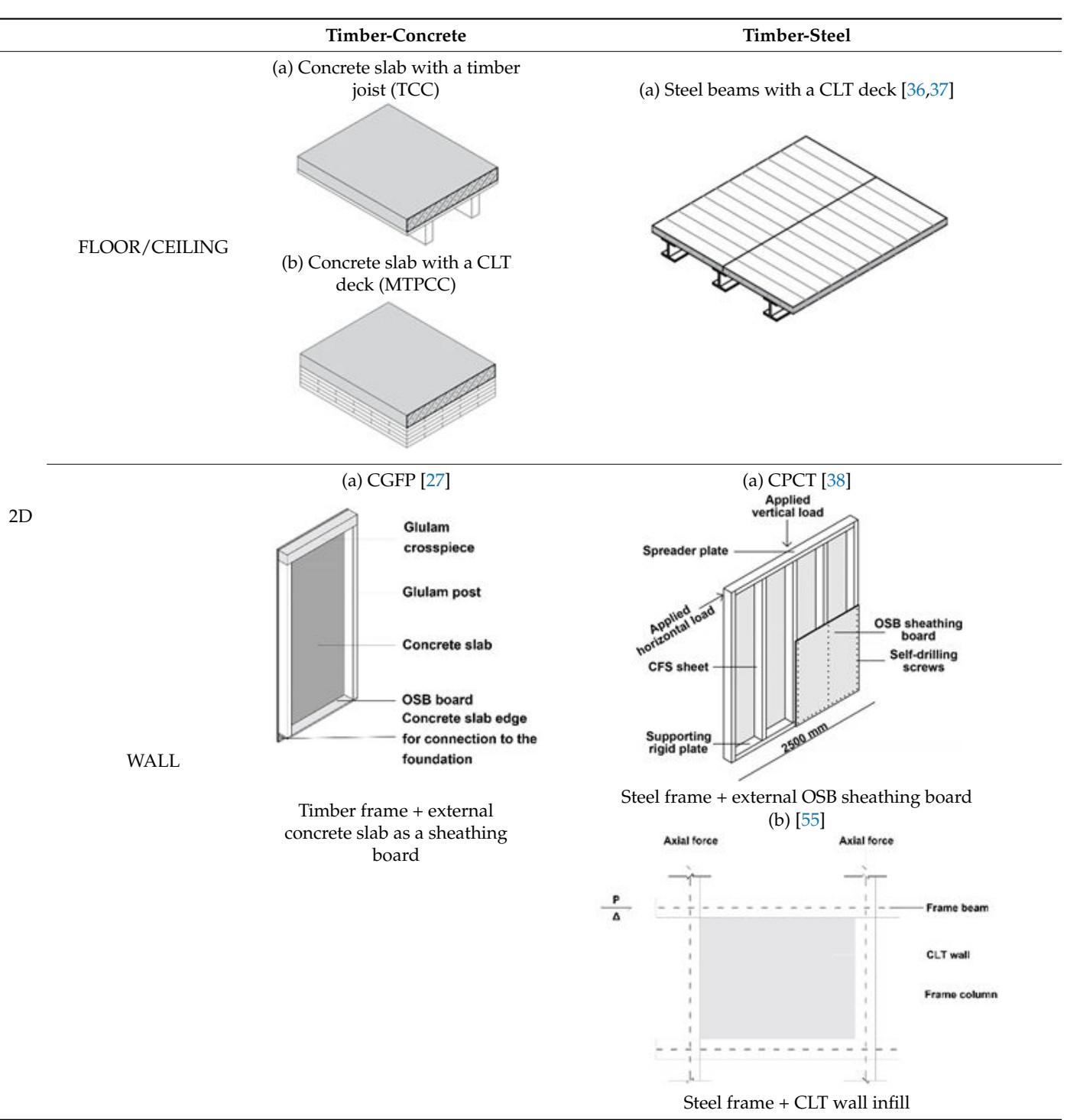

| | Timber-Concrete | Timber-Steel |
|---|---|---|
| FLOOR/CEILING | (a) Concrete slab with a timber joist (TCC)<br><br>(b) Concrete slab with a CLT deck (MTPCC) | (a) Steel beams with a CLT deck [36,37] |
| 2D<br><br>WALL | (a) CGFP [27]<br><br>Glulam crosspiece<br>Glulam post<br>Concrete slab<br>OSB board<br>Concrete slab edge for connection to the foundation<br><br>Timber frame + external concrete slab as a sheathing board | (a) CPCT [38]<br>Applied vertical load<br>Spreader plate<br>Applied horizontal load<br>CFS sheet<br>OSB sheathing board<br>Self-drilling screws<br>Supporting rigid plate<br>2500 mm<br><br>Steel frame + external OSB sheathing board<br>(b) [55]<br>Axial force / Axial force<br>P Δ<br>Frame beam<br>CLT wall<br>Frame column<br><br>Steel frame + CLT wall infill |

The examples listed in Table 7 are explained in more detail below.

(a)    Hybrid timber-concrete structural elements/components (TCC)

Timber-Concrete Floor Composites (TCC)

As the weaknesses of prefabricated timber floor systems are low bending stiffness, high sensitivity to pedestrian vibrations, and also relatively low horizontal stiffness to assure the horizontal structural diaphragms mechanism to be able to transfer horizontal load impact to the load-resisting wall elements, solutions of strengthening timber floor elements

with another material are usually crucial. In view of the latter and to simultaneously meet various criteria, i.e., structural and building physics criteria, hybrid timber-concrete composite (TCC) structural systems have been more frequently used with one structural timber system applied to floor elements. The first applications of TCC floor elements were related to the refurbishment process of existing old timber floors. In the past 50 years, the interest in TCC floors has rapidly increased, as well as in the construction of new buildings [50]. Various studies of lightweight timber floor elements [42,75] state that for example, in Italy, the TCC approach is often rejected by the authorities responsible for the cultural heritage preservation due to insufficient reversibility. Compared with the previously presented CLT solutions of strengthening prefabricated concrete elements, combinations with timber joists are better from many aspects, but special consideration of connectors is required. In this sense, a rigid timber-to-concrete connection is the most effective solution for timber-concrete composite members subjected to the flexure that provides full composite action and better structural behaviour. One of the mostly used technologies to produce a glued connection of the timber-concrete composite is the "dry" method, which includes the gluing of timber and precast concrete slab together, as proposed in [28]. On the other hand, the advantages of the TCC approach compared with a CLT slab are better sound insulation properties [76] and better vibrational behaviours [51]. Furthermore, the TCC and CLT strengthening approach of lightweight timber floors is desired to have the lowest possible height to minimally change the existing floor levels and to ensure the room functionality. In [53], the use of additional gaps by placing timber blocks between the CLT panel and timber joists shows how these gaps increase the composite cross-section, resulting in higher stiffness and the strength of the floor. Most currently available studies of TCC floor elements are focused on simply supported slabs, as this is a typical configuration of timber buildings. However, in other structural applications and for reinforced concrete buildings, the boundary conditions of TCC slabs are not likely to be simply supported. Therefore, an analytical procedure for designing timber-concrete composites (TCC) subjected to various boundary conditions other than simply supported is presented in a study [29].

There are some environmental impacts of building materials if alternative CLT or concrete (TCC) solutions for hybrid floor elements solutions are compared, which must be additionally considered among only the structural results discussed above. For objective comparisons between CLT and TCC, the equivalent structural performance (bending resistance) has to be set as the main fixed boundary condition. The LCA study [30], in which the Korean building codes and standards are used, states that the CLT slab emitted 75% less carbon dioxide in comparison with the TCC solution. A similar study [31] explores the life-cycle greenhouse gas emissions (LCGHGE) and life-cycle primary energy (LCPE) of three high-rise residential buildings in a cold region of China through a life-cycle assessment approach. The three buildings are conventional reinforced concrete (RC), CLT, and hybrid CLT buildings. The results show that CLT and hybrid CLT buildings produce 15.00% and 10.77% lower LCGHGE, respectively, compared to the RC building within the 50-year service life. Research on the environmental impact of multi-storey timber buildings is briefly presented in [45] with further special attention on the influence of steel fasteners on the LCA results. The results presented as relative and absolute contributions of different building elements show that the embodied impacts per floor decrease as the building height increases.

Certain additional strengthening techniques have also been performed to increase the bending resistance of existing old timber floors. The study in [19], in which timber joists are additionally strengthened with glass strips and using an RC slab on the top of the floor element, is not only the first holistic approach to evaluate the structural and environmental performances of the proposed strengthening technique, but also addresses the aesthetically-aware design and technical limitations of the utilisation of glass for the renovation of timber floors and thoroughly presents the possibilities to overcome these limitations. Further information about the structural and environmental behaviours of

TCC elements and all advantages and disadvantages can be found in the state-of-the-art reviews about TCC floor elements [26,32,50].

Mass Timber Panel-Concrete Composite (MTPCC)

Wider spans of timber-based floor elements sometimes require the use of timber composite floor elements with a CLT panel instead of lightweight timber joists at the bottom in combination with an RC slab on the top. Mass timber panel-concrete composite (MTPCC) floors combine a top concrete layer with bottom mass timber panels, such as cross-laminated timber (CLT). It has been found, for example, in [33] that the maximum composite efficiency in a mass timber panel-concrete composite floor (MTPCC) can be achieved by optimising the number, size, and locations of notches, and the maximum bending stiffness of the floor can be achieved without increasing the cost or self-weight of the floor system.

Timber-Concrete Wall Composites

In the field of composite timber-concrete wall composites, there are less structural solutions than those for composite timber-concrete floor elements. Only three such solutions have been found in the available literature. It is important to point out that a hybrid combination of timber-based structures with concrete wall elements will also improve the thermal capacity and acoustic performance of a building. The study in [27] presents an innovative solution with the development of the timber-concrete prefabricated composite wall system, i.e., the concrete glulam framed panel (CGFP), which is a panel made of a concrete slab (sheathing) and a structural glulam frame. The results of the presented analysis significantly improve the structural and thermal behaviours of a building. The developed composite wall elements can be used as strengthening bracing elements in prefabricated tall timber buildings, especially when exposed to heavy wind or seismic load. In the developed composite timber-concrete wall system [34], conventional and commonly used sheathing boards (OSB or fibre-plaster boards) were replaced with thin concrete panels. The interaction between the timber frame and a thin reinforced concrete slab connected to the timber frame panel has been experimentally identified in order to update the finite element model and simulate the structural performance of such a composite wall element. Another detailed study of a composite load-bearing timber-concrete wall element composed of a timber frame and a thin external reinforced concrete board is presented in [77]. The timber frame structure transfers the vertical load to the foundation, while the bracing system reacting to horizontal actions consisting of OSB panels and three square reinforced concrete boards connected to the wood frame.

(b)　Hybrid timber-steel structural elements/components (STC)

To improve ductility and consequently the overall seismic resistance of timber structures, recent research has focused on the development of hybrid timber-steel structures, instead of relying on all-timber structures only [78–81]. Such hybrid structures can provide a significant dissipative capacity if designed with adequate strength, stiffness, and ductility [20]. In comparison with the already presented hybrid timber-concrete structural composite solutions, the mass of the whole structure in this case is evidently smaller, and the ductility is better, which is essential for significantly better seismic resistance of timber-based multi-storey buildings. As presented in Table 7 for one-dimensional (1D) linear load-bearing elements (columns and beams), a steel element is usually inserted into the timber element, which protects the steel element against corrosion and fire exposure. In such a composite load-bearing structural system, a steel element is usually primarily responsible for increasing the structural resistance, stiffness, and especially the ductility of the structural component. However, timber elements usually protect steel against climate impacts and fire exposure. In some cases, especially if such composite 1D or 2D floor elements (such as by the composite floor solutions [36,37]) are used only in a completely dry climate and indoor conditions, the steel element can also be placed outside of the timber

element, without any climate or fire protection by timber. However, the fire resistance of steel components in such cases must be achieved by special fire protection coatings.

Timber-Steel Floor Composites

The vibration behaviour of steel-timber composite floors (STC) was experimentally and numerically studied in the research [39]. It was demonstrated that, in comparison with the timber-concrete composite (TCC) floor components presented above with a lower weight of STC, STC beams also satisfied all vibration requirements for floors ($f_1 > 8$ Hz), according to the Eurocode 5 standard. Therefore, STC can be justified as a suitable alternative solution according to TCC floor components, especially if there are some limitations to the additional maximal possible weight, which can be adopted for structural renovation of old existing timber floors. There are still some disadvantages in terms of worse thermal capacity, as it can be better achieved with TCC renovation solutions.

Timber-Steel Wall Composites

The timber-steel composite wall element solution with a newly developed prefabricated load-bearing closed composite timber-steel wall panel (CPCT) made of oriented strand boards (OSB) stiffened by sawn-cut timber stud and sometimes with an additional steel stud to increase its load-carrying capacity was proposed in a study [21]. However, this research proposes the utilisation of a diaphragm to transfer the gravity load only. However, lateral loads were not analysed. The study in [38] developed a composite wall panel composed of oriented strand boards (OSB) and a cold-formed steel frame. An OSB board is connected to the steel frame outside of the steel frame as a sheathing material. Such a lightweight composite system can be proposed for the construction of wall and floor systems in low-rise multi-storey buildings within the framework of a rapidly urbanising society. A similar hybrid steel-timber wall element was developed in [56], which is composed of an internal frame realised with tubular steel columns and timber beams, which is externally braced with OSB panels on both sides and fastened to the frame with proper dowel-type fasteners. The new building system is an evolution of the one tested and described in [58]. Because of increased ductility, the incorporation of steel columns within OSB bracing panels results in a strong and stiff platform frame system with a high potential in low- and medium-rise buildings in seismic areas. Unfortunately, the wall elements are not load-resisting enough for high-rise timber-based buildings, especially for those located in seismically more active areas. Thus, further study on such a steel–timber hybrid shear wall (STHSW) system in terms of better energy dissipation was carried out in [57]. In this study, a new system, i.e., self-centering (SC)-STHSW, is proposed by introducing post-tensioned (PT) technology into the previously developed STHSW system. Many parametrically numerical analyses were performed, and based on the obtained results, a design parameter, i.e., a self-centering ratio, was proposed. This study provides important support for the application of the innovative steel–timber hybrid structural wall system to be further used in cases of practical engineering.

A different structural approach to composite timber-steel wall elements applicable also to taller hybrid timber buildings with essentially higher lateral resistance of wall elements was first used in [49]. In this study, a timber-steel composite lateral load-resisting wall element was formed using a combination of a ductile steel frame and a low-ductile CLT deck instead of OSB sheathing. It is important to point out that, in this case, the CLT wall panel is connected to the steel frame inside the frame as an infill and not externally as in previously described timber-steel wall models. The developed steel–timber hybrid wall system was further numerically designed as a parametric study of the lateral load-resisting system of a three-, six-, and nine-storey timber-based building for the seismicity of Vancouver with two different ductility levels (ductile and limited ductility) considered. As for many other similarly developed timber-based composite beam and wall structural elements, it is of the utmost importance to accurately evaluate the connection stiffness influence and, specifically in this case, the contribution of the infill CLT wall panel to the

stiffness and strength of such a hybrid wall system. The load-sharing effect between the CLT wall and the steel frame was experimentally and numerically studied in [55]. The numerical results showed that the connected models were very effective, as the CLT infill absorbed a substantial part of the lateral load during the initial stages of loading.

4.2.2. Examples of Erected Tall Buildings in Hybrid Timber-Based Structural Systems

To improve the seismic behaviour of mid-rise and high-rise timber buildings, various solutions were proposed using a reinforced concrete (RC) structural core. Especially if the transparent glass areas are asymmetrically placed on the building envelope, the RC core can essentially decrease torsion effects (twisting) caused by the horizontal load impact in each storey and increase an overall lateral resistance of the whole building. A good example of this is the 24-storey and 84-metre-high HoHo (German abbreviation from Holz-Hochhaus) Tower in Vienna completed in 2019 and recognised at that time as the highest timber building in Europe. The structural system is a hybrid timber structure with a concrete core, where the staircase, lift, and technical shafts are located. Concrete cores improve the lateral resistance of the whole structure [82,83].

Another example is the Haut building, a 73-metre-tall hybrid timber-concrete building in Amsterdam. It is currently regarded as the third tallest timber building in Europe and the highest in the Netherlands [84]. The multi-storey timber building consists of 2 underground floors and 21 floors above ground, with the ground and first floor made of concrete, while the other 19 floors are built of timber with one RC core [85]. A compact high-rise tower made of wood in a slightly irregular and asymmetrical polyhedral form is equipped with projecting glass façades. Because of the asymmetrical position of the lateral load-bearing wall elements, heavy torsional effects can appear especially due to a seismic load action, which is very low (0.05 g) [60]. It is estimated that the $CO_2$ emissions of the Haut building compared to an all-concrete version are 34 tonnes instead of 870 tonnes [86].

It is characteristic for both described hybrid timber-concrete buildings that, due to their extreme height and exposure to the strong wind impact load, they cannot be constructed only in a timber load-resisting structural braced system like the previously described 18-storey Mjøstårnet building in Brumunddal. Therefore, special reinforced concrete (RC) cores are required in both cases. There is a special desire to construct them manually with eco-friendly materials while respecting the ecological conditions. Therefore, timber, in both cases, is used as a primary building material (in about 70% or more) and concrete is used only to improve the structural properties, especially the lateral stability and the resistance of prefabricated floor elements.

However, there are also some new recently developed structural solutions in hybrid timber-concrete structural systems, which significantly improve the structural behaviours of tall timber-based buildings. For example, a study in [54] presented a hybrid timber-concrete building composed of two parts: a concrete core with concrete flat slabs on every third floor as the main structure, and prefabricated light timber frame modules as substructures. The authors also explore the newly developed hybrid system feasibility by comparing a 30-storey hybrid timber-concrete building with a traditional all-concrete alternative at a site in Vancouver. The results show that the seismic mass of the proposed hybrid system is reduced when compared to the concrete building, resulting in a shorter fundamental period and lower seismic load demand [26].

Selected examples of tall timber buildings have been presented to demonstrate the application of the structural systems in practical examples. More information on the tall timber buildings and their structural characteristics can be found in [59,61,70].

## 5. Summary of the Results and Discussion

To conclude the description of individual timber structural systems, Table 8 provides a summary of some structural properties of individual structural systems and timber-based hybrid systems, which affect the selection of the appropriate type of construction. The characteristics are additionally presented according to the load-bearing structural

limitations with the maximal possible number of storeys with the selected bracing systems, the maximal floor-span, and with carbon sequestration caused by the type of the structural system.

**Table 8.** Some main structural characteristics of all-timber and timber-based hybrid structural systems.

| | Timber Structural Systems | | | | | | |
|---|---|---|---|---|---|---|---|
| | All-Timber | | | Hybrid Timber-Based | | | |
| Type | Solid Timber (CLT) | Frame | Frame-Panel (LTF) | Timber-Timber | Timber-Concrete | Timber-Steel | Timber-Concrete Steel |
| Approx. max. number of storeys | 10 | 14 | 4 | 18 | 24 | 9 | / |
| Approx. max. horizontal span (m) | 9.0 | 6.0 | 6.0 | 10.0 | 15.0 | 10.0 | 15.0 |
| Carbon sequestration (low–high) | High | Medium | Medium | Medium/ high | Low | Low | Low |
| Case studies (already erected buildings) | Forte (Melbourne) | Treet (Bergen, Norway) | Many 4-storey buildings | Mjøstårnet (Brumunddal, Norway) | HoHo Tower (Vienna, Austria) | / | / |
| Bracing system in the case study building | 5-layer CLT elements | Glulam frame elements | Timber frame + sheathing boards | Combination of glulam frame and CLT | Combination of glulam frame and RC core | Combination of timber and steel frame | / |

Many numerical studies showed that the lateral resistance, especially the lateral stiffness, and consequently the seismic resistance of LTF wall elements (frame-panel structural system) are not particularly high, but on the other hand, the U-value of the external envelope wall elements and the price are the lowest. For example, the study in [22] performed fragility analysis of LTF shear walls, while the study in [23] investigated damage to LTF wall elements due to an earthquake. The study in [24] developed a framework for the loss estimation of timber construction subjected to seismic loads. Due to relatively low lateral resistance and lateral stiffness of LTF wall elements [25], there are still some limitations to the height and number of storeys of timber buildings (maximum four storeys) constructed in the light timber-framed (LTF) wall system, especially of buildings erected in heavily windy or seismically active areas. Therefore, the solid timber (CLT) wall load-resisting system or the frame (skeletal) system is required instead of the LTF structural system for timber buildings with more than four storeys. To assess the lateral resistance of LTF and CLT prefabricated wall elements, numerical analyses were performed, which analysed and compared both types of prefabricated wall elements [7,43]. It was concluded that the lateral resistance of CLT wall elements is evidently higher than that of LTF wall elements with the same wall dimensions. Therefore, most mid-rise (more than three storeys) timber buildings are constructed in practice with CLT wall load-resisting elements instead of LTF elements to satisfy the ultimate limit state and serviceability limit state prescriptions set in [74]. On the other hand, taking into account the aspects of energy demand regarding the U-values of the envelope elements, it is more appropriate to use LTF wall elements instead of CLT, since with equal thickness of the wall, lower U-values can be obtained with LTF. However, the maximum height of prefabricated buildings with the CLT structural wall system is ten storeys (Forte building, Melbourne). In the case of taller buildings, the skeletal frame system with glulam linear load-bearing frame elements is usually required

to decrease the weight and consequently the seismic action on the whole building. Using such a single structural load-resisting bracing system, buildings with up to 14 storeys have already been constructed in practice (Treet building, Bergen in Norway). However, because of essentially greater wind or seismic actions, buildings higher than 14 storeys require the use of the hybrid all-timber structural bracing system instead of the single load-resisting timber system. In this case, usually a hybrid combination of the CLT core and the frame envelope structural system is used. The CLT core is assumed to sustain most of the lateral forces acting on the building. On the other hand, a frame envelope structure assumes the optimal U-value of external wall elements and an additional part of lateral resistance. Such an example of all-timber hybrid construction is the 18-storey Mjøstårnet building in Brumunddal, Norway.

Very tall timber buildings (more than 18 storeys) require a combination of the timber lateral load-resisting system and another structural load-bearing material due to greater wind or seismic actions. Reinforced concrete (RC) is mainly used, which also facilitates an increase in the thermal capacity of the whole building and an improvement in the energy efficiency of the building. However, in such a case, the carbon sequestration of the building is essentially decreased compared with a possible alternative with an all-timber load-bearing structure. A concrete structure as a strengthening construction material is mostly used as the RC core, optimally located in the centre of the building to sustain most of the greater horizontal load actions on the structure, and to decrease the floor-plan irregularity of the whole building. An example is the 24-storey HoHo Tower in Vienna, 74% of which is made of wood. Thanks to this approach, the construction saves 2800 t $CO_2$ equivalent compared to a conventionally constructed RC building of the same type and size [82].

## 6. Conclusions

Due to its many ecological advantages, there is a growing trend towards timber construction worldwide, which is increasingly extending to multi-storey construction of both medium-rise (four to ten storeys) and high-rise timber buildings (more than ten storeys). The choice of the most appropriate structural system is mainly determined by the height of the building, and its exposure to wind or seismic loads. However, due to the very low modulus of elasticity and also the relatively low thermal capacity, there are a number of limitations to timber construction, in particular in the case of high-rise timber structures.

The main objective of this paper was to provide a systematic overview of the existing contemporary timber construction systems and their main features, due to the lack of scientific literature providing such a comprehensive overview. The latter presents a novelty in the body of knowledge of timber construction review. It might also provide designers with an overview of all existing timber structural systems and their specificities to make the right selection of a proper contemporary solution and to support the design process.

The results of the review show that in practice, the tallest timber building erected exclusively in one load-bearing system (massive-panel) to date is a 10-storey building, and the tallest in a hybrid timber construction as a combination of two timber construction systems is an 18-storey building. In the case of even taller timber buildings (more than 18 storeys), the most optimal solution is usually the choice of a hybrid construction system, where the primary load-bearing timber structure is combined with load-bearing elements made of other materials. The function of these additional materials is to serve as structural reinforcement elements that increase the primary horizontal load-bearing capacity of the building, thus enabling greater heights of buildings to be achieved. In this case, the timber structure is usually combined with reinforced concrete, which also greatly increases the thermal storage capacity of the building, but also considerably increases the overall mass of the building. Still, if the added mass is to be reduced and, in particular, the ductility of the structure to be increased, the most appropriate solution is to add a steel superstructure. Nevertheless, timber is still the primary material for the load-bearing structure and facilitates the best possible environmental performance. Such buildings are

referred to as hybrid timber-based structures. So far, the highest such structure is a 24-storey structure. However, new design solutions are likely to lead to at least a 30-storey hybrid timber structure very soon.

In conclusion, there is still some potential for advances in the timber construction technology, which can lead to even more sophisticated and resilient solutions in the field of timber buildings.

**Author Contributions:** Conceptualisation, M.P. and V.Ž.L.; methodology, M.P. and V.Ž.L.; writing—original draft preparation, M.P. and V.Ž.L.; writing—review and editing, M.P. and V.Ž.L. All authors have read and agreed to the published version of the manuscript.

**Funding:** Funding for this research was partly provided by the Slovenian Research Agency, National research programme P2-0129.

**Acknowledgments:** The authors would like to thank Maja Lešnik Nedelko for administrative support in the preparation of the images of timber structural systems and elements.

**Conflicts of Interest:** The authors declare no conflict of interest. The funders had no role in the design of this study; in the collection, analyses, or interpretation of data; in the writing of the manuscript; or in the decision to publish the results.

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
