# Peer review of "Innovative Structural Systems for Timber Buildings: A Comprehensive Review of Contemporary Solutions"

_buildings, doi:10.3390/buildings13071820_

Round 1

Reviewer 1 Report

The authors provide an overview of the construction systems for timber buildings based on a systematic review of the scientific literature. The purpose does not appear! Not even in the introduction. After this overview, what is new in this study?

Abstract: good overview of the problem and definition of the research purpose, it is advisable to emphasise more the results obtained and especially the novelty of the research, which appears little. How can this study be useful for academics and researchers? Avoid similarities with summaries of articles that already exist in the literature.

Subsection 1.1. "Systematic literature review", concerns the methodological approach, so it needs its own section and not a subparagraph of the introduction. The methodology is an important section of the scientific paper, it needs to be clear and straightforward in order to be replicable in further studies and to facilitate the understanding of the approach chosen for writing the text. It must be given in the text and not only in Table 1.
It is necessary to include the bibliographic references of the PRISMA method (lines 75-76). Explain the reason for the selection of the bibliographic databases. Why were the databases listed in lines 76-77 selected? Scopus is an official database for scientific literature, while the other two databases mentioned are restricted to some publishers. Why this choice? How was the database search conducted? What keywords were used to obtain the 43585 results. Was a search by abstract-keywords-title or by other criteria conducted?

Drastically revise your study methodology. Each step of the systematic review shown in Figure 1 needs to be described in detail to explain the decisions made from the identification phase to the screening phase. How were the inclusion and exclusion criteria determined? How did the screening phase take place? It must be specified in the text and not just presented in a table. The inclusion criteria show that only articles from the year 2000 onwards were selected. Why this selection? This must be specified in the text. It is not clear from the inclusion criteria that "selected words are included in the title keywords and abstract". Please clarify this criterion and in particular clarify why title keywords and abstract fall under the inclusion criteria and are not the database search method (as they should be). Authors are advised to read the PRISMA methodological guidelines carefully.
The papers go from 43585 (data identification phase) to 58 (data inclusion phase), but the screening phase is not clear.
Clarify the content of the text from line 90 to 106? If this does not fall under the prism method, it is advisable to create a separate subsection.
The prism method would also require that the meta-analysis phase provide quantitative results, but the authors do not provide any information in this regard.
After applying the PRISMA method, the results should come from the articles selected in the "included data" phase. Are these results reported in section 2? I seem to read a new paragraph that does not refer to the analysis of the articles. Where are the references to the selected articles?

After the detailed description of the different structural/constructive systems made of wood, the original and innovative contribution of this research is not clear. Separate the results from the discussions and conclusions, not as in the current form, but by highlighting the difference between these paragraphs and especially the novelty of the study.

Please review the structure of some sentences and avoid some repetitions, to improve the reading and fluency of the text.

Author Response

Dear Reviewer 1,

we thank you for your time and constructive comments that helped us improve the quality of this manuscript. In the attachment you can find the document, we have responded to your comments. 

Kind regards, Authors

Reviewer 2 Report

Dear authors

This review paper summarized the contemporary wood buildings including all-timber and hybrid timber structural systems, which contributes to this field of research and applications. However, some issues should be made to improve the paper.

1) It is recommended that review the commonly wood species and wood-based materials used in timber building and compare the applicability of these materials. That will more interesting to readers. 

2) Meanwhile, it is recommended that the famous wood buildings and their information should be summarized.

3) In line 49-50, it is said that "Its compressive strength is almost equal to that of concrete, but its tensile strength is significantly higher". The reference proofs this point must be added like"Experimental and numerical studies on mechanical behaviors of beech wood under compressive and tensile states "

4) In line 67-68, Just as you mentioned, why the joints of wood construction were not covered in this paper? As known that joint is the critical point which determines the strength of the whole structure.

5) In section 1.1, It is recommended that change this subtitle as "Methodology of literature research ".

6) For table 2, How did you make the criteria you selected confident and reliable. These criterion made you miss some important studies. It is recommended that some non-English writing but with English abstract should be also considered.

7) Also in section 1.1, It is recommended that use a figure to show the keywords distributions of all literature included in this study. 

8) In section 2, The subtitle is not numbered correct. Only 2.1 was found, there is no 2.2 found.

9) Also, some mistakes were found in the text, and texts were in bold. Please double check them and modify. 

Author Response

Dear Reviewer 2,

we thank you for your time and constructive comments that helped us improve the quality of this manuscript. In the attachment you can find the document, we have responded to your comments. 

Kind regards, Authors

Round 2

Reviewer 1 Report

The efforts of the authors are commendable, who have succeeded in a very short time (perhaps even too little) in revising the manuscript and clarifying its novelty in comparison with existing studies in the literature.
Unfortunately, the methodology still has serious gaps in the implementation of the PRISMA method.
The previous version of the paper reported 43585 identified papers; the revised version reports 4532. What is the true figure?
The authors write in line 120 that 4532 records were identified taking into account the inclusion criteria, but the records in the "data identification" phase are those resulting from searching the databases without applying the inclusion criteria. The inclusion criteria must be applied in the "data screening" phase to screen the results to answer the research questions, with the aim of achieving the objectives of the work.
It sounds strange and puzzling that 4050 out of 4532 papers are not accessible in the full text version as stated by the authors. Previously, the authors had claimed that they preferred a particular literature database precisely because it was freely accessible.
The authors do not state the reason for limiting the search to articles published from the 2000s onwards, and also ignore the meta-analysis that accompanies the prism method to provide a quantitative analysis of the results.

Author Response

Deaar Reviewer 1,

thank you for your time and consideration of our paper.

However, we need to clatify once again the issues regarding literature search methodology.

As already mentioned in the first answer to your comments, in the first literature search (in originally submitted paper), we used the first filter without limitation to Title-Abstract-Keyword, especially for Science Direct. However, in the second step which is also shown as initial step in the revised paper, we used this filter, which drastically reduced the number of articles. This was also explained in the answers to the comments of Reviewer 1, where we explain this mistake. However, for all search databases, we did not set the first filter to "fully accessible articles", but due to our access rights to university articles, we were only able to check this criterion after the first search. Also, some of these articles showed a  lack of relevance to the topic after abstract review (but this was not decisive, since they were not accessible in full text to be read). The figure preceding the inclusion criteriia clearly states that these are the criteria that apply to the final selection of papers, so that the data are true and justified. As for the speed of the corrections of the review, which you  reproach us for, this is a matter of conditions on the part of the journal (review within 10 days). However, since we had the initial search information recorded in our notes, it was possible to make the corrections in such a short time.   And regarding your comment: It sounds strange and puzzling that 4050 out of 4532 papers are not accessible in the full text version as stated by the authors. Previously, the authors had claimed that they preferred a particular literature database precisely because it was freely accessible.   It is written in the answer to your comment and in the paper that we have selected DOAJ, since it is freely accesible, and not ScienceDirect or Scopus.   Regarding the limitation to 2000 - it is since there is hardly any literature on COONTEMPORARY systems being published prior 2000. There ae many reasons, why literature on modern systems, which includes composite elements, high rise buildings, has been started to appear after 2000. It is a matter of technology development, which we are trying to present in the current paper. The authoruty of the authors who are familiar with the field is the decission on the appropriate time interval and not every decision has to be explained in the paper text.    However, our goal was to present the sistematic overwiew of the existing technology in the field of timber structural systems. We are recgnized experts in this field and even if not conductiong this researc with a PRISMA method, we are familiar with new trends and development  in timber construction, so the way to the selected literature might be different, but is not the main point. of the articlle, the main novelty is a systematic presentation of all new systems - state of the art of currenttly developed technology.  I hope, this can be understood from the paper.   I also hope, this answer satisfies your concern on issues explained.

Reviewer 2 Report

Dear authors

Although some of issues were not addressed appropriately, the paper was much improved. I recommended that it can be accepted for publishing.

Author Response

Dear Reviewer 2,

thank you for your time and cnsideration of our paper.

Round 3

Reviewer 1 Report

Well done!

Author Response

Dear Reviewer 1,

thank you for your time and cnsideration of our paper.